# Real-Time Tracheal Ultrasound vs. Capnography for Intubation Confirmation during CPR Wearing a Powered Air-Purifying Respirator in COVID-19 Era

**DOI:** 10.3390/diagnostics14020225

**Published:** 2024-01-21

**Authors:** Seungwan Eun, Hee Yoon, Soo Yeon Kang, Ik Joon Jo, Sejin Heo, Hansol Chang, Guntak Lee, Jong Eun Park, Taerim Kim, Se Uk Lee, Sung Yeon Hwang, Sun-Young Baek

**Affiliations:** 1Department of Emergency Medicine, Samsung Medical Center, Sungkyunkwan University School of Medicine, Seoul 06355, Republic of Koreaikjoon.jo@samsung.com (I.J.J.); sejin.heo@samsung.com (S.H.); hansol.chang@samsung.com (H.C.); guntak.lee@samsung.com (G.L.); jongeun7.park@samsung.com (J.E.P.); taerimi.kim@samsung.com (T.K.);; 2Department of Emergency Medicine, Chung-Ang University Gwangmyeong Hospital, Gwangmyeong-si 14353, Republic of Korea; ifely@naver.com; 3Biomedical Statistics Center, Data Science Research Institute, Research Institute for Future Medicine, Samsung Medical Center, Seoul 06351, Republic of Korea; sun0.baek@sbri.co.kr

**Keywords:** cardiopulmonary resuscitation, endotracheal intubation, ultrasound, personal protective equipment, CPR, emergency department: point-of-care ultrasound

## Abstract

This study aimed to compare the accuracy of real-time trans-tracheal ultrasound (TTUS) with capnography to confirm intubation in cardiopulmonary resuscitation (CPR) while wearing a powered air-purifying respirator (PAPR). This setting reflects increased caution due to contagious diseases. This single-center, prospective, comparative study enrolled patients requiring CPR while wearing a PAPR who visited the emergency department of a tertiary medical center from December 2020 to August 2022. A physician performed the TTUS in real time and recorded the tube placement assessment. Another healthcare provider attached waveform capnography to the tube and recorded end-tidal carbon dioxide (EtCO_2_) after five ventilations. The accuracy and agreement of both methods compared with direct laryngoscopic visualization of tube placement, and the time taken by both methods was evaluated. Thirty-three patients with cardiac arrest were analyzed. TTUS confirmed tube placement with 100% accuracy, sensitivity, and specificity, whereas capnography demonstrated 97% accuracy, 96.8% sensitivity, and 100% specificity. The Kappa values for TTUS and capnography compared to direct visualization were 1.0 and 0.7843, respectively. EtCO_2_ was measured in 45 (37–59) seconds (median (interquartile range)), whereas TTUS required only 12 (8–23) seconds, indicating that TTUS was significantly faster (*p* < 0.001). No significant correlation was found between the physician’s TTUS proficiency and image acquisition time. This study demonstrated that TTUS is more accurate and faster than EtCO_2_ measurement for confirming endotracheal tube placement during CPR, particularly in the context of PAPR usage in pandemic conditions.

## 1. Introduction

Endotracheal intubation is necessary to secure and maintain the airway in patients presenting to the emergency department (ED) with respiratory failure, altered mentality, airway obstruction, or cardiac arrest. It is crucial to ensure that the endotracheal tube (ETT) is properly positioned in the trachea, as accidental tube misplacement during emergency intubation can have severe adverse consequences [1]. Conventional methods for confirming tube placement include direct visualization of the tube within the larynx, the presence of water vapor within the tube, symmetrical chest rise, auscultation of lung sounds, and chest X-rays. In addition, end-tidal carbon dioxide (EtCO_2_) measurement is regarded as one of the most reliable methods owing to its high accuracy and ease of application [2,3,4,5].

In the case of endotracheal intubation during cardiopulmonary resuscitation (CPR), however, many of the aforementioned methods have limitations [6,7]. The sensitivity of EtCO_2_ is significantly lower in patients with cardiac arrest, at 67.9%, compared to 98.8% in non-arrest patients [7,8,9,10]. During the COVID-19 pandemic, it is recommended for physicians performing intubations to wear personal protective equipment (PPE), including powered air-purifying respirators (PAPRs). This requirement not only makes auscultation impractical but also could impair vision due to the PAPR hood. Consequently, this may increase the complexity of the intubation process [11,12]. In addition, using mechanical compression devices such as load-distributing band type complicates the problem, making auscultation or visualization of chest wall rising impossible.

In CPR and other resuscitation scenarios, ultrasound has become an increasingly vital tool for rapid assessment and decision-making [13,14,15]. Additionally, it is widely recognized as a powerful screening and diagnostic tool in the ED, particularly for evaluating tracheal intubation. Trans-tracheal ultrasound (TTUS) demonstrates high specificity and sensitivity for tube confirmation, and its learning curve is relatively short, requiring only one to two practice cases to achieve the proficiency reported in previous studies [16,17]. Ultrasound is also useful for differentiating between tracheal and bronchial intubation by confirming bilateral lung sliding. Furthermore, it affords a great advantage in the setting of CPR using a mechanical compressor while wearing a PAPR, where inspection of bilateral elevation of the chest wall or auscultation of both lung fields is difficult to conduct.

Some studies have compared the accuracy of ultrasound and capnography in confirming tube placement in a cardiac arrest setting [1,18], but no studies have compared these techniques in CPR environments wearing PAPRs. Therefore, this research investigates the effectiveness of ultrasound for verifying ETT placement in CPR situations involving the use of PAPRs, filling a previously unexplored gap in existing studies.

## 2. Methods

### 2.1. Study Design and Setting

This research was a single-center, prospective, comparative study carried out in the ED of a tertiary university-affiliated medical center in South Korea. The medical center is notable for its high patient volume, handling approximately 70,000 patients annually. The study spanned from December 2020 to August 2022, during a period marked by significant healthcare challenges globally. This study’s experimental protocol, involving human participants, was meticulously designed and conducted in strict adherence to the ethical principles outlined in the Declaration of Helsinki. Approval for the study was obtained from the Samsung Medical Center Institutional Review Board (IRB), underlining the study’s compliance with the required ethical and procedural standards. The IRB granted a waiver of consent for this study, recognizing the nature of the research and the conditions under which it was conducted (IRB file number 2020-11-115-004). Furthermore, this study was officially registered with ClinicalTrials.gov (Date 30 December 2020, ID NCT04690517).

### 2.2. Participants

The inclusion criteria for this study required participants to be patients aged 18 years or older who experienced cardiac arrest and underwent intubation while ED medical staff wore PAPRs. The mandatory use of PAPRs was a precaution against the elevated risk of transmitting contagious diseases, which was particularly relevant in the clinical setting and due to the nature of the involved procedures.

Exclusion criteria were established based on the feasibility of performing TTUS. Patients with head and neck trauma were excluded, as such injuries could complicate or contraindicate TTUS use. Similarly, patients with anatomical deformities caused by previous neck surgeries or the presence of neck masses were not included, considering the potential impact of these conditions on the accuracy and practicability of TTUS. Additionally, this study excluded patients for whom capnography could not be administered during intubation. Since capnography was a critical component of the comparative analysis of this study, its applicability was essential for inclusion. This comprehensive approach to inclusion and exclusion criteria ensured that this study focused on a suitable patient population, enabling a reliable assessment of TTUS in comparison to capnography for intubation verification in emergency care.

### 2.3. Methods and Measurements

#### 2.3.1. Study Protocol

In response to the challenges posed by the COVID-19 pandemic, our study protocol was meticulously designed to address the unique circumstances surrounding the treatment of cardiac arrest patients with potential infectious diseases. When emergency medical services dispatch a call indicating the arrival of such patients at the hospital, medical staff are required to don PAPRs for the duration of CPR. This protocol is in line with enhanced safety measures to protect healthcare workers from airborne pathogens while providing critical care. In addition to these precautions, all equipment necessary for intubation, including the ultrasound machine and EtCO_2_ measurement devices, was pre-arranged and readily available in the resuscitation room.

Upon patient arrival, high-quality CPR was started immediately according to the American Heart Association (AHA) guidelines for advanced cardiac life support (ACLS) [19]. For this purpose, AutoPulse^®^ Resuscitation System (Zoll Medical Corporation, Chelmsford, MA, USA), an automated chest compressor device, was utilized to ensure consistent and effective chest compressions. Endotracheal intubation, a critical advanced airway strategy, was then performed. The choice between using a direct laryngoscope or a video-assisted laryngoscope for this procedure was left to the discretion of the intubating physician, based on their clinical judgment and preference.

During the intubation, to maintain the integrity and unbiased approach of the study, separate and independent medical professionals were responsible for the subsequent procedures. A different physician conducted the real-time TTUS to assess the placement of the tube, while another healthcare provider, not involved in the intubation process, attached waveform capnography to the tube and recorded the EtCO_2_ level after 5 breaths. The study protocol involved comparing the accuracy of tube placement judgment using both ultrasound and waveform capnography, using direct visualization of the tube through the glottis as the gold standard. Furthermore, the timing for the evaluation of both the TTUS and EtCO_2_ began from the moment the ETT was inserted into the oral cavity. We meticulously recorded the time required to confirm tube placement using each method. This approach provided valuable data on the efficiency of these methods. After tracheal intubation, a lung ultrasound was performed on the upper regions of both lungs.

#### 2.3.2. Waveform Capnography

A Capnostream™ 35 Portable Respiratory Monitor (Medtronics, Minneapolis, MN, USA) was used to measure EtCO_2_. The device was attached to the endotracheal tube, and a continuous waveform and EtCO_2_ level were shown on the display. EtCO_2_ values were recorded after 5 breaths. An EtCO_2_ value lower than 4 was considered esophageal intubation, and a value equal to or greater than 4 was considered endotracheal intubation [19].

#### 2.3.3. Ultrasound

A brief educational session was provided to every participating emergency physician taking the role of the sonographer. The sessions consisted of lectures on basic anatomy, as seen on TTUS, ultrasound findings for tracheal and esophageal intubation, and a hands-on program for TTUS and lung ultrasound (Figure 1, Appendix A).

Two ultrasound devices were used in this study: HM70A (Samsung Medison, Seoul, Republic of Korea) and Vivid S70N (GE Healthcare, Chicago, IL, USA). Seven to twelve megahertz high-frequency linear transducers were used for the image acquisition. The probe was placed in a transverse plane 2 cm above the suprasternal notch of the patient during the intubation process in real time. The hyperechoic tracheal ring and posterior reverberation artifact forming the “comet tail sign” were considered endotracheal intubations. Esophageal intubation is identified by the presence of hyperechoic rings with posterior reverberation artifacts in the esophagus, resulting in the formation of two hyperechoic “double ring signs”- one in the trachea and one in the esophagus [20]. In the absence of a “double ring sign” in TTUS, endotracheal intubation was presumed even if the intra-tracheal “comet tail sign” could not be ensured. To evaluate the correct position of the tube after intubation, bilateral lung sliding in the upper zone of both lungs was assessed during CPR [19,21]. If lung sliding was observed on only one side of the lung field, the tube was withdrawn 2–3 cm, and a lung ultrasound was performed again for re-evaluation.

#### 2.3.4. Data Collection

Data collected from the enrolled patients included age, sex, body mass index, and medical history. The results and time required for assessing tube placement using tracheal ultrasound and waveform capnography were collected. After confirming tracheal intubation, sliding in both lungs was evaluated. In addition, CPR and intubation-related information were collected: no/low flow time, CPR time in the ED, result of CPR, type of laryngoscope, anticipated difficult airway, and first-pass success rate.

### 2.4. Outcomes

The primary objective of this study was to identify the accuracy of TTUS and capnography in confirming appropriate tube placement compared with direct visualization. The level of agreement between TTUS and capnography, relative to this gold standard, was quantified using the Kappa value. The secondary objective was to compare the differences in the time to confirm tube placement between capnography and TTUS. In addition, whether the proficiency of the sonographer influences the accuracy and time in TTUS confirmation of ETT placement was investigated.

### 2.5. Sample Size and Statistical Analysis

The target number of participants was assumed using data from previous studies comparing the accuracy of TTUS and capnography upon intubation during CPR [1,7]. The accuracy of capnography was 0.65, and the accuracy of TTUS was expected to be 0.95. The expected ratio for the mismatch between capnography and gold standards and the match between TTUS and gold standards was 0.34, and the assumed ratio for a match between capnography and gold standards but not between TTUS and gold standards was 0.04. To test the primary hypothesis using McNemar’s test with a statistical significance of 5% and power of 80%, the required number of participants was 31. The expected drop-out was set at 10%; thus, a minimum of 34 participants were required to participate in this study.

Continuous variables are expressed as mean and standard deviation (SD) or median and interquartile range (IQR). Categorical variables are presented as numbers and percentages. The accuracy between the gold standard and TTUS and capnography values was analyzed using McNemar’s test, and the agreement of TTUS and capnography compared to the gold standard was expressed by the Kappa value. The Kappa value was further classified into four groups according to the guidelines suggested by Koo and Li et al. (2016): 0–0.50 as poor agreement, 0.50–0.75 as moderate agreement, 0.75–0.90 as good agreement, 0.90–1.00 as excellent agreement [22]. Comparison of time taken to confirm ETT placement with TTUS and capnography was tested with the Wilcoxon signed rank sum test. The correlation between sonographer proficiency and time to TTUS confirmation was analyzed using the Kruskal–Wallis and Jonckheere–Terpstra trend tests. Statistical analysis for the study outcomes was performed using PASS2020 v20.0.2 and SAS version 9.4 (SAS Institute Inc., Cary, NC, USA).

## 3. Results

A total of 322 patients with cardiac arrest underwent CPR in the ED during the study period. Patients who were already intubated (*n* = 35), intubated after the return of spontaneous circulation (ROSC) (*n* = 15), had never been intubated (*n* = 35), received CPR without PAPR (*n* = 102), did not apply EtCO_2_ (*n* = 6) or TTUS (*n* = 92), and had missing data (*n* = 2) were excluded. One case in which EtCO_2_ could not be measured and one case in which TTUS was performed after ROSC dropped out, and a total of thirty-three patients were analyzed (Figure 2).

There were 24 males (73%), and the mean (standard deviation, SD) age of the enrolled participants was 70 (15) years. The mean (SD) body mass index was 23 (3). Most cardiac arrests occurred outside the hospital (OHCA) (32 out of 33 cases), and the median (interquartile range, IQR) values of no-flow time (time from onset of cardiac arrest to initiation of basic life support (BLS)) and low-flow time (time from initiation of BLS to initiation of ACLS) were 8 (0–19) min and 29 (22–36) min, respectively. Thirty-three percent of the patients had ROSC, and only one survived more than 24 h. All chest compressions of CPR were performed using a mechanical compressor, and all intubations, except one, were performed through video laryngoscopy. The median percentage of glottic opening (POGO) score was 100, and a difficult airway was anticipated in five cases (15%) (Table 1).

Capnography was used to evaluate tube placement, of which thirty patients were considered endotracheal, and three were esophageal. Its accuracy compared with direct visualization was 97%, with a sensitivity of 96.8% and a specificity of 100%. Through TTUS, the insertion of endotracheal tubes was observed in real time, and tube placement was confirmed as endotracheal and esophageal in thirty-one and two patients, respectively. This result was completely consistent with a direct laryngoscopic view, with an accuracy of 100%. Capnography showed good agreement with a Kappa value of 0.78, and TTUS showed excellent agreement with a Kappa value of 1.0 [22].

The time (median (IQR)) to acquire the EtCO_2_ value was 45 (37–58.5) s, while the TTUS approach required 12 (8–23) s, with TTUS being significantly faster (*p* < 0.001) (Table 2, Figure 3). There was no statistically significant correlation between the TTUS proficiency of the physician and the image acquisition time (Appendix A). Lung ultrasound was performed after confirming the tube inside the trachea in twenty-six cases (79%), and inspection of the bilateral upper lung field revealed one case of unilateral lung sliding, indicating bronchial intubation.

## 4. Discussion

Confirmation of tracheal tube placement is important, particularly in the COVID-19 era. Falsely placed tubes harm not only the patient but also expose healthcare providers to the potential risk of a contagious disease [23]. Wearing PPE by medical staff, common in the COVID-19 era, creates various physical restrictions and raises the necessity to verify whether the methods used to confirm tube placement still function while wearing PPE. This is the first study to determine tracheal intubation with TTUS while wearing a PAPR during CPR for cardiac arrest patients. Although the number of participants was modest, TTUS was demonstrated to be a very useful procedure with 100% accuracy for tube confirmation. Moreover, the median time for confirming intubation by TTUS was only 12 s, which was significantly faster than the time to EtCO_2_ measurement of 45 s. Consequently, the performance of TTUS was as good as that of studies in which medical staff did not wear PPE [1,24]; it is considered a useful approach to confirm intubation during CPR while wearing PAPRs during the pandemic era.

Ultrasound has several advantages that make it suitable for cardiac arrest and critical care settings. First, real-time ultrasound assessment of tube passing does not require cessation of chest compression [1]. Thus, tube placement confirmation using ultrasound in addition to clinical assessment is strongly recommended as an alternative when waveform capnography is unavailable in the 2020 ACLS guidelines [21]. Lung ultrasound after intubation can also identify the main bronchus intubation, which is not feasible with capnography. Even though the sonographic window was limited to the upper lung fields by the compressor band of the automated CPR machine applied in this study, one right main bronchus intubation was identified, and the tube depth was successfully adjusted. In addition, combining various sonographic views makes ultrasound an even stronger tool to help manage cardiac arrest patients by identifying reversible causes and determining the presence or absence of a pulse [24].

Another advantage of TTUS is the short learning curve. According to one study, it was possible to confirm accurate tube placement with only a brief online tutorial and two practice attempts [17]. In a study on paramedics, tube confirmation could be effectively taught with only a short, simulation-based training session [25]. However, image acquisition is much more difficult during CPR because the patient constantly moves as chest compression progresses. In this study, even physician novices to TTUS with less than five practice experiences could perform the procedure and interpret results without significant difficulty. Statistical analysis demonstrated that the number of experiences with TTUS correlated with neither higher accuracy nor shorter time taken for tube placement confirmation. Although there have been no studies on the precise learning curve of TTUS, a brief hands-on session is sufficient for proficiency.

Continuous waveform capnography is recommended in addition to clinical assessment as the most reliable method of confirming and monitoring the correct placement of an endotracheal tube according to the ACLS guidelines of AHA [19,21]. Nevertheless, there are studies reporting low sensitivity and false negatives in circumstances where carbon dioxide production is decreased (i.e., pulmonary embolism or low cardiac output state) [7,9,10,24,26]. However, the accuracy, sensitivity, and specificity of capnography in this study were all greater than 95%, compared to 60 to 65% in previous investigations of cardiac arrest. Only one patient with a prolonged low-flow time had an EtCO_2_ value of 0 despite endotracheal intubation, but there were no false-positive cases in the study. Contrary to previous research, capnography appeared to be beneficial in the context of cardiac arrest. In addition, it provides information regarding the quality or termination of CPR. However, the risk of large-volume aspiration of gastric contents and rapid hypoxia due to erroneous insertion into the esophagus must always be considered, as capnography can confirm the result only after ventilation through the tube.

PPE is recommended in patients suspected of having an airborne infection during aerosol-producing procedures, such as endotracheal intubation. However, PAPRs may distract medical staff and limit their physical activities. One study showed that 2 min shift chest compressions result in a significantly lower sufficient chest compression rate compared to the 1 min shift chest compressions group during CPR while wearing PAPRs [27]. In studies regarding endotracheal intubation, intubation failure rate or airway complications do not increase with PPE use [28,29], but medical staff complained of significant discomfort [30]. Particularly, it may be challenging to evaluate the location of the tube because medical personnel cannot undertake a physical examination or auscultation while wearing PAPRs. Under these circumstances, TTUS provides fast and accurate results, making it an essential intubation confirmation tool when there is a risk of infectious disease transmission.

### Limitations

Our study has some limitations. First, it was conducted at a single institution with a small sample size, potentially limiting the generalizability of our findings to a broader patient population. Additionally, the study included only a few cases of difficult airways, and esophageal intubation was performed in just two patients. This limited occurrence raises the possibility of a type 2 error, making it challenging to extend these findings to a wide range of airway situations. Second, we used convenience sampling due to restrictions on the number of medical staff involved in CPR, aimed at minimizing COVID-19 transmission. This method might have introduced a selection bias, especially since ultrasound could not be performed in some cases due to delays in preparing the equipment and the rapid nature of the intubation process. Third, our study did not account for the potential impact of mechanical CPR, particularly with a load-distributed band device like AutoPulse™, on EtCO_2_ measurements. This aspect should be considered when interpreting our findings on EtCO_2_ evaluation efficiency. Fourth, Multiple methods should be used to confirm proper endotracheal tube placement, as using two or more independent tests together improves overall sensitivity. However, this study did not evaluate several commonly used clinical signs. Finally, the non-storage of ultrasound images presents a significant limitation, as it hinders the assessment of image quality and inter-rater reliability. This lack of image data analysis could affect the interpretation and reproducibility of our results. Together, these factors underscore the need for larger-scale, multi-center studies to validate our findings and enhance the understanding of ultrasound’s role in airway management during CPR.

## 5. Conclusions

This study demonstrates that TTUS is a more effective method than capnography for confirming endotracheal tube placement during CPR, particularly when healthcare providers are equipped with PAPRs in pandemic situations. TTUS not only boasts a 100% accuracy rate but also significantly outperforms capnography in terms of speed, with a median time of 12 s for TTUS compared to 45 s for capnography. Additionally, the proficiency of the physician in TTUS did not significantly affect image acquisition time, underscoring TTUS’s effectiveness irrespective of operator expertise. The findings suggest that incorporating TTUS could substantially improve emergency airway management, potentially enhancing patient outcomes in urgent care scenarios.

## Figures and Tables

**Figure 1 diagnostics-14-00225-f001:**
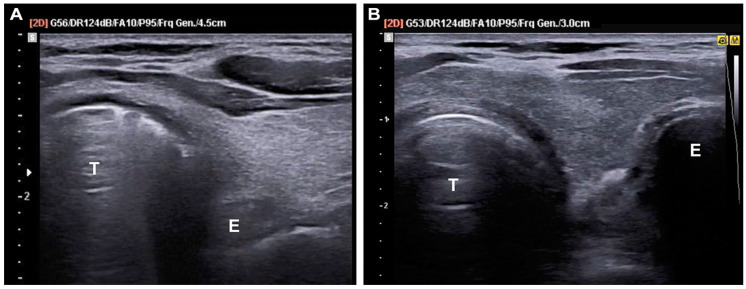
Trans-tracheal ultrasound images during endotracheal intubation. (**A**) “Comet tail sign” indicating endotracheal intubation; (**B**) “Double ring sign” indicating esophageal intubation. T, trachea; E, esophagus.

**Figure 2 diagnostics-14-00225-f002:**
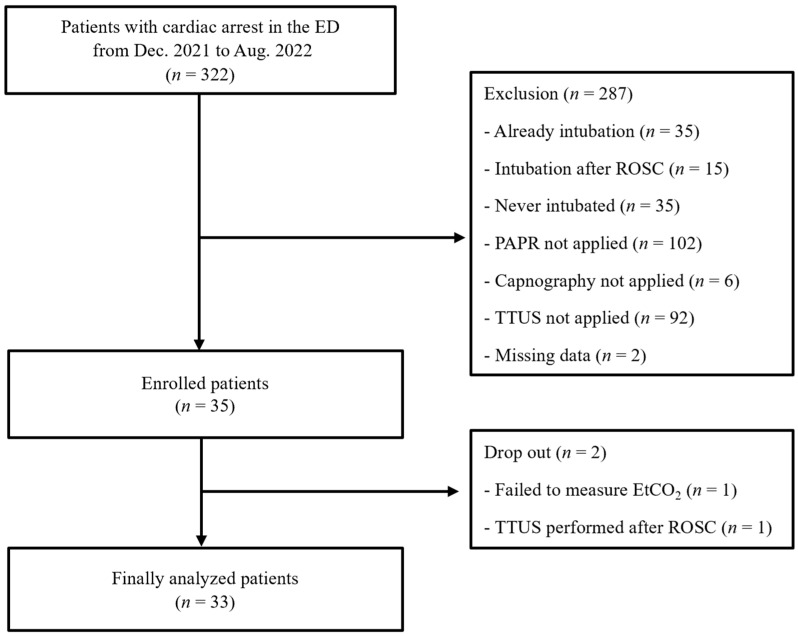
Study flow diagram. ED, emergency department; PAPR, powered air-purifying respirator; TTUS, trans-tracheal ultrasound; ROSC, return of spontaneous circulation.

**Figure 3 diagnostics-14-00225-f003:**
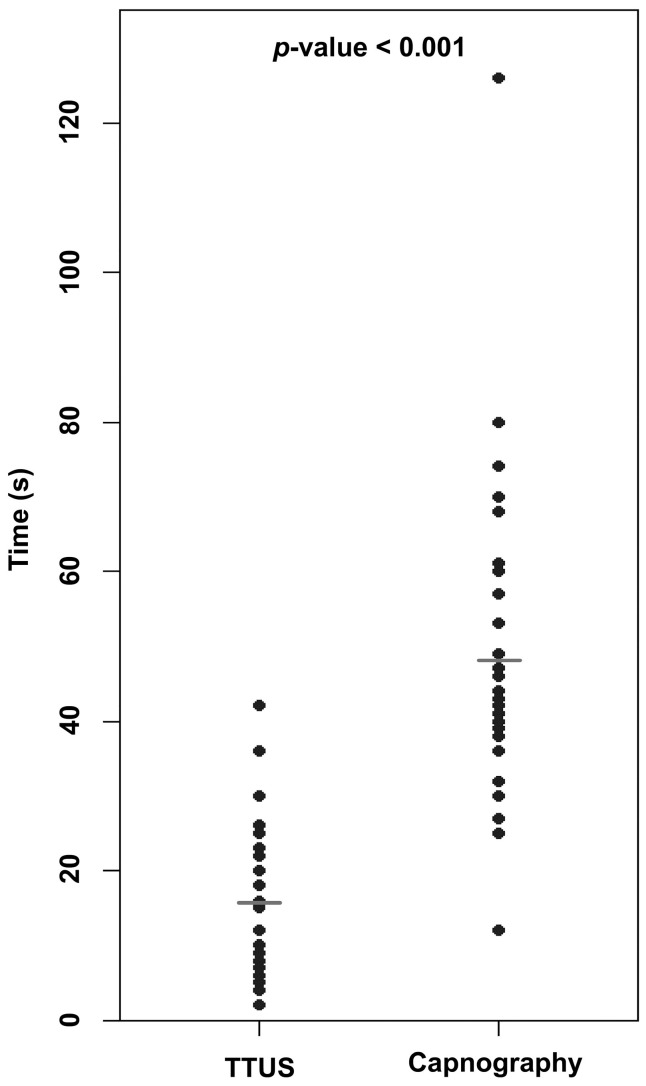
Median time spent to confirm tube placement by capnography and TTUS. TTUS, trans-tracheal ultrasound.

**Table 1 diagnostics-14-00225-t001:** Demographic and clinical characteristics of patients.

Characteristics	Descriptives (*n* = 33)	Characteristics	Descriptives (*n* = 33)
**General Characteristics**		Respiratory	11 (34%)
Sex (male)	24 (73%)	Non-traumatic bleeding	2 (6%)
Age (years)	70 ± 15	Unknown	10 (30%)
BMI (kg/m^2^)	23 ± 3	ROSC	
**Underlying Diseases**		No ROSC	22 (67%)
Hypertension	17 (52%)	ROSC, death < 24 h	10 (30%)
Diabetes mellitus	21 (67%)	ROSC, death > 24 h	1 (3%)
Cardiac diseases	26 (79%)	**Intubation Evaluation**	
Chronic lung diseases	27 (82%)	Type of laryngoscopy	
Chronic liver diseases	29 (88%)	Direct laryngoscopy	1 (3%)
Chronic renal diseases	28 (85%)	Video laryngoscopy	32 (97%)
Cerebral vascular accident	28 (85%)	O_2_ therapy before intubation	
Malignancy	28 (85%)	LMA	6 (18%)
COVID-19 positive	1 (3%)	BVM	25 (76%)
**Characteristics of CPR**		ET tube	2 (6%)
Type of cardiac arrest		Difficult airway anticipated	
OHCA	32 (97%)	POGO	100 (75–100)
IHCA	1 (3%)	GOG	
No-flow time (min)	8 (0–19)	1	29 (88%)
Low-flow time (min)	29 (22–36)	2	4 (12%)
ACLS time (min)	21 ± 8	3, 4	0
Cause of cardiac arrest		First pass success	29 (88%)
Traumatic	3 (9%)	Tube depth (cm)	23 (23–25)
Cardiogenic	7 (21%)	Complication	0

Data are reported as numbers (percentage, %), mean (standard deviation, SD), or median (interquartile range, IQR). BMI, body mass index; COVID-19, coronavirus disease-2019; CPR, cardiopulmonary resuscitation; OHCA, out-of-hospital cardiac arrest; IHCA, in-hospital cardiac arrest; ACLS, advanced cardiac life support; ROSC, return of spontaneous circulation; LMA, laryngeal mask airway; BVM, bag–valve mask; ET tube, endotracheal tube; POGO, percentage of glottis opening; GOG, glottis opening grade.

**Table 2 diagnostics-14-00225-t002:** Comparison of capnography and TTUS methods for ETT placement confirmation.

	TTUS	Capnography
Tracheal intubation (*n* = 31)	31	30
Esophageal intubation (*n* = 2)	2	3
Time spent to confirm tube placement, (s) (median, IQR) *	12 (8–22)	45 (37–59)
Sensitivity, % (95% CI)	100 (88.8–100)	96.8 (83.3–99.9)
Specificity, % (95% CI)	100 (15.8–100)	100 (15.8–100)
PPV, % (95% CI)	100	100
NPV, % (95% CI)	100	66.7 (22.5–93.2)
Accuracy, % (95% CI)	100 (89.4–100)	97.0 (84.2–99.9)
McNemar test (*p*-value)	N/A	0.317
Kappa	1.00 (1.00–1.00)	0.78 (0.38–1.00)

* The time difference using the Wilcoxon signed-rank test is statistically significant (*p*-value < 0.001). IQR, interquartile range; CI, confidence interval; PPV, positive predictive value; NPV, negative predictive value; TTUS, transtracheal ultrasound; ETT, endotracheal tube.

## Data Availability

The datasets used and/or analyzed in the current study are available from the corresponding author upon reasonable request.

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
