# Peer review of "Real-Time Tracheal Ultrasound vs. Capnography for Intubation Confirmation during CPR Wearing a Powered Air-Purifying Respirator in COVID-19 Era"

_diagnostics, 2024, doi:10.3390/diagnostics14020225_

Round 1

Reviewer 1 Report

Comments and Suggestions for Authors

Thank you for the opportunity to review this article on the use of ultrasound for intubation confirmation.

Before I started reading this article, I did a search on PubMed and found about 200 articles from 2010 onward on this topic, which means that the deal of novelty of this paper  is not that high.

Anycase, the paper is well written and  readable even though I found some criticisms:

1) It is not clear to me who performed echo. Was the same rescuer who performed intubation? You stated the ETCO2 sensor was attached by another rescuer. What about echo? In my view if they who performed intubation also confirmed it with echo they may have been biased by laringoscopy and by the view of the tube passing through the vocal cords. Please specify this pivotal point.

2) Probably there is something wrong with the number of patient. 30+15+35+102+6+92+2 = 282 and not 287. As a consequence the patients enrolled should be 40 and not 35 and the final population should have made up of 38 patients and not 33. Please verify. 

3) You did not mention the causes underlying the cardiac arrests included in your study. The presence of causes generally associated to low values of ETCO2 may have influenced the results. Please provide the reader with this information.

4) It is not clear how you assessed the time of execution of echo and of ETCO2 evaluation. Did you consider strictly the time needed to obtain and interpret the image and the ETCO2 curve or did you consider the time needed to have the equipment ready? You say that echo was faster but probably the ultrasound machine takes more time to be ready that the monitor for ETCO2 which is usually turned on at the beginning of ACLS for ECG analysis and defibrillation.

5) I would organise abstract  with the standard paragraph: introduction, methods, results, conclusions.

6) You might mention the fact that ETCO2 curve may be affected by mechanical CPR with AutoPulse to enrich your discussion[Resuscitation. 2016 Jun;103:e9-e10. doi: 10.1016/j.resuscitation.2016.03.003.]

Author Response

Reviewer 1

Thank you for the opportunity to review this article on the use of ultrasound for intubation confirmation.

Before I started reading this article, I did a search on PubMed and found about 200 articles from 2010 onward on this topic, which means that the deal of novelty of this paper is not that high.

Anycase, the paper is well written and readable even though I found some criticisms:

>> Thank you for your detailed review and insights on our manuscript, which explores the application of ultrasound for intubation confirmation in the COVID-19 era. Your constructive feedback is instrumental in refining our paper and enhancing its contribution to the field.

Our study specifically addresses the unprecedented challenges brought forth by the COVID-19 pandemic, focusing on the treatment of cardiac arrest patients potentially suffering from infectious diseases. The pandemic has necessitated the use of Powered Air-Purifying Respirators (PAPR) by medical staff during CPR, significantly impacting traditional intubation methods. Our research stands out in this landscape by investigating the use of trans-tracheal ultrasound (TTUS) under these unique circumstances. We aimed to rigorously evaluate the utility of TTUS in confirming endotracheal tube placement during CPR, considering the complexities such as impaired vision due to PAPR hoods and the infeasibility of auscultation. This study contributes a critical perspective to the field, specifically addressing the challenges of ensuring accurate tube placement in the constrained and visually limited conditions of PAPR use during the pandemic.

We revised the manuscript for emphasizing this point.

The title has been updated as follows:

"Real-time Tracheal Ultrasound vs. Capnography for Intubation Confirmation during CPR with a Powered Air-Purifying Respirator in the COVID-19 Era".

Some sentences in the abstract have been modified as follows:
“Introduction: This study aimed to compare the accuracy of real-time trans-tracheal ultrasound (TTUS) with capnography to confirm intubation in cardiopulmonary resuscitation (CPR) while wearing a powered air-purifying respirator (PAPR). This setting reflects increased caution due to contagious diseases.

Conclusion: This study demonstrated that TTUS is more accurate and faster than EtCO2 measurement for confirming endotracheal tube placement during CPR, particularly in the context of PAPR usage in pandemic conditions.”

We revised some sentences in the introduction as follows (Lines 61-66) (Lines 86-88):
“During the COVID-19 pandemic, it is recommended for physicians performing intubations to wear personal protective equipment (PPE), including powered air-purifying respirators (PAPR). This requirement not only makes auscultation impractical but also could impair vision due to the PAPR hood. Consequently, this may increase the complexity of the intubation process”

“Therefore, this research investigates the effectiveness of ultrasound for verifying ETT placement in CPR situations involving the use of PAPR, filling a previously unexplored gap in existing studies.

We revised some sentences in the conclusion as follows (Lines 426-429)

“This study demonstrates that TTUS is a more effective method than capnography for confirming endotracheal tube placement during CPR, particularly when healthcare providers are equipped with PAPRs in pandemic situations.”

C1) It is not clear to me who performed echo. Was the same rescuer who performed intubation? You stated the ETCO2 sensor was attached by another rescuer. What about echo? In my view if they who performed intubation also confirmed it with echo they may have been biased by laringoscopy and by the view of the tube passing through the vocal cords. Please specify this pivotal point.

A1>> Thank you for your valuable comments. We revised the manuscript that emphasizes the independence of the medical professionals involved in intubation, TTUS, and EtCO2 measurement as follows (Lines 152-158):

“During the intubation, to maintain the integrity and unbiased approach of the study, separate and independent medical professionals were responsible for the subsequent procedures. A different physician conducted the real-time TTUS to assess the placement of the tube, while another healthcare provider, not involved in the intubation process, attached waveform capnography to the tube and recorded the EtCO2 level after 5 breaths.”

C2) Probably there is something wrong with the number of patient. 30+15+35+102+6+92+2 = 282 and not 287. As a consequence the patients enrolled should be 40 and not 35 and the final population should have made up of 38 patients and not 33. Please verify.

A2>> We reviewed all the data, and we found that there were 35 patients who were already intubated. Thank you for finding the error, I corrected it and recreated Fig. 2.  

And we also revised the manuscript. (Line 255)

“Patients who were already intubated (n = 35), intubated after return of spontaneous circulation (ROSC) (n = 15), had never been intubated (n = 35), received CPR without PAPR (n = 102), did not apply EtCO2 (n = 6) or TTUS (n = 92), and had missing data (n = 2) were excluded.”

C3) You did not mention the causes underlying the cardiac arrests included in your study. The presence of causes generally associated to low values of EtCO2 may have influenced the results. Please provide the reader with this information.

A3>> We have reviewed the data and categorized the causes of cardiac arrest, incorporating this information into Table 1. The analysis revealed that respiratory arrest was the predominant cause, accounting for 34%, followed by unknown causes at 30%. This detailed breakdown of arrest causes is crucial as it may influence the interpretation of EtCO2 values and their implications in the study results. This addition enriches the understanding of the cardiac arrest patients involved in our study. Thank you for your insightful comment, which has not only helped in enhancing the study's depth but also provides the reader with critical context.

Table 1 (Line 279)

Characteristics

Descriptives

(n = 33)

Characteristics

Descriptives

(n = 33)

General Characteristics

 Respiratory

11 (34%)

Sex (male)

24 (73%)

 Non-traumatic bleeding  

2 (6%)

Age (years)

70 ± 15

 Unknown

10 (30%)

BMI (kg/m2)

23 ± 3

ROSC

Underlying Diseases

No ROSC

22 (67%)

Hypertension

17 (52%)

ROSC, death < 24 h

10 (30%)

Diabetes mellitus

21 (67%)

 ROSC, death > 24 h

1 (3%)

Cardiac diseases

26 (79%)

Intubation Evaluation

Chronic lung diseases

27 (82%)

Type of laryngoscopy

Chronic Liver diseases

29 (88%)

 Direct laryngoscopy

1 (3%)

Chronic renal diseases

28 (85%)

 Video laryngoscopy

32 (97%)

Cerebral vascular accident

28 (85%)

O2 therapy before intubation

Malignancy

28 (85%)

 LMA

6 (18%)

COVID-19 Positive

1 (3%)

 BVM

24 (73%)

Characteristics of CPR

 ET tube

2 (6%)

Type of cardiac arrest

Difficult airway anticipated

 OHCA

32 (97%)

POGO

100 [75–100]

 IHCA

1 (3%)

GOG

No flow time (minutes)

8 [0–19]

1

29 (88%)

Low flow time (minutes)

29 [22–36]

 2

4 (12%)

ACLS time (minutes)

21 ± 8

  3, 4

0

Cause of cardiac arrest

First pass success

29 (88%)

Traumatic

3 (9%)

Tube depth (cm)

23 [23–25]

 Cardiogenic

7 (21%)

Complication

0

C4) It is not clear how you assessed the time of execution of echo and of EtCO2 evaluation. Did you consider strictly the time needed to obtain and interpret the image and the EtCO2 curve or did you consider the time needed to have the equipment ready? You say that echo was faster but probably the ultrasound machine takes more time to be ready that the monitor for EtCO2 which is usually turned on at the beginning of ACLS for ECG analysis and defibrillation.

A4>> In response to your question about how we assessed the execution time of trans-tracheal ultrasound (TTUS) and EtCO2 evaluation, I would like to provide some clarification on our methodology. Both TTUS and EtCO2 equipment were prepared in advance in the resuscitation room. The ultrasound machine was turned on and ready upon receiving notification from emergency medical services about an incoming cardiac arrest patient. The intubation equipment and EtCO2 devices were also pre-arranged. However, cases where measurements could not be made because ultrasound machines and capnography were not available were excluded from the study, and this is described in limitations. The time measurement for both TTUS and EtCO2 evaluations began from the moment the E-tube was inserted into the oral cavity. We recorded the time taken to confirm tube placement using each method. For EtCO2, we used a portable device attached to the E-TUBE, and the EtCO2 values were recorded after 5 ventilations.

We added the following sentences to clarify the prepared setting in advance. (Lines 138-141)

“In addition to these precautions, all equipment necessary for intubation, including the ultrasound machine and ETCO2 measurement devices, was pre-arranged and readily available in the resuscitation room.”

In addition, we clarified how we assessed the time of execution of TTUS and EtCO2 evaluation as follows (Lines 161-164).

“Furthermore, the timing for the evaluation of both the TTUS and EtCO2 began from the moment the ETT was inserted into the oral cavity. We meticulously recorded the time required to confirm tube placement using each method. This approach provided valuable data on the efficiency of these methods.”

C5) I would organise abstract with the standard paragraph: introduction, methods, results, conclusions.

A5>> We revised the abstract as follows:

“Introduction: This study aimed to compare the accuracy of real-time trans-tracheal ultrasound (TTUS) with capnography to confirm intubation in cardiopulmonary resuscitation (CPR) while wearing a powered air-purifying respirator (PAPR). This setting reflects increased caution due to contagious diseases.

Methods: The single-center, prospective, comparative study enrolled patients requiring CPR while wearing a PAPR who visited the emergency department of a tertiary medical center from December 2020 to August 2022. A physician performed the TTUS in real time and recorded the tube placement assessment. Another healthcare provider attached waveform capnography to the tube and recorded end-tidal carbon dioxide (EtCO2) after five ventilations. The accuracy and agreement of both methods compared with direct laryngoscopic visualization of tube placement, and the time taken by both methods were evaluated.

Results: Thirty-three patients with cardiac arrest were analyzed. TTUS confirmed tube placement with 100% accuracy, sensitivity, and specificity, whereas capnography demonstrated 97% accuracy, 96.8% sensitivity, and 100% specificity. The Kappa value for TTUS and capnography compared to direct visualization were 1.0 and 0.7843, respectively. EtCO2 was measured in 45 (37–59) seconds (median [interquartile range]), whereas TTUS required only 12 (8–23) seconds, indicating that TTUS was significantly faster (P < 0.001). No significant correlation was found between the physician's TTUS proficiency and image acquisition time.

Conclusion: This study demonstrated that TTUS is more accurate and faster than EtCO2 measurement for confirming endotracheal tube placement during CPR, particularly in the context of PAPR usage in pandemic conditions.”

C6) You might mention the fact that EtCO2 curve may be affected by mechanical CPR with AutoPulse to enrich your discussion [Resuscitation. 2016 Jun;103:e9-e10. doi: 10.1016/j.resuscitation.2016.03.003.]

A6>> The 2016 "Resuscitation" article highlights the impact of mechanical CPR, especially using devices like AutoPulse™, on EtCO2 readings. It points out that mechanical CPR creates minimal tidal volume, affecting EtCO2 numerical values, which tend to be low and fluctuating. However, the EtCO2 waveform curve remains stable and reliable, leading to a recommendation to prioritize the waveform for more accurate CPR assessments.

In our study, instead of continuous monitoring EtCO2 devices, we utilized a portable capnography device (Capnostream™ 35 Portable Respiratory Monitor, Medtronics, Minneapolis, MN) connected to the E-tube for simple value measurements. This meant that we could not observe and evaluate the EtCO2 waveform curve. Recognizing this as a limitation, we will include in our study that the absence of continuous waveform analysis could potentially affect the interpretation of EtCO2 values, especially under the influence of LBD type mechanical CPR as suggested by the referenced article in 'Resuscitation' (2016).

We revised the limitation section by adding the following sentences (lines 410-414).

“Third, our study did not account for the potential impact of mechanical CPR, particularly with a load-distributed band device like AutoPulse™, on EtCO2 measurements. This aspect should be considered when interpreting our findings on EtCO2 evaluation efficiency.

We appreciate the opportunity to resubmit our work and hope that it now meets the journal's standards for publication.

Reviewer 2 Report

Comments and Suggestions for Authors

The article should specify what is new about what is already known. The use of ultrasound for rapid intubation is a commonly used technique. The article makes no reference to the benefit of the ultrasound technique over others that may be useful, such as direct video laryngoscopy. 

The authors should justify their study with respect to others in the literature. 

Once the changes have been made, the text could be re-evaluated for publication.

Author Response

Reviewer 2

The article should specify what is new about what is already known. The use of ultrasound for rapid intubation is a commonly used technique. The article makes no reference to the benefit of the ultrasound technique over others that may be useful, such as direct video laryngoscopy.

The authors should justify their study with respect to others in the literature.

Once the changes have been made, the text could be re-evaluated for publication.

>> Thank you for your valuable review of our manuscript concerning the utilization of ultrasound for intubation confirmation. In response to the challenges presented by the COVID-19 pandemic, we meticulously devised our study protocol to address the unique circumstances surrounding the treatment of cardiac arrest patients with potential infectious diseases.

When emergency medical services dispatch a call indicating the arrival of such patients at the hospital, medical staff are mandated to don powered air-purifying respirators (PAPR) for the duration of CPR. This protocol aligns with enhanced safety measures to shield healthcare workers from airborne pathogens while delivering critical care.

The use of PPE, specifically PAPR, complicates traditional methods such as auscultation, rendering it impractical. Visual cues like chest wall rise are often obscured, especially when utilizing mechanical compression devices like load-distributing bands. The physical restrictions imposed by PPE on medical staff underscore the need to verify whether the methods employed for confirming tube placement remain effective while wearing PPE.

Our study endeavors to underscore the effectiveness and utility of trans-tracheal ultrasound (TTUS) under these specific conditions, providing a reliable alternative for confirming tube placement when direct visualization and auscultation are impeded. Intubation while wearing a PAPR can impair vision due to the PAPR hood, making it challenging for someone other than the operator to provide supervision in a narrow airway treatment space. Consequently, this may elevate the complexity of the intubation process. In these scenarios, TTUS serves as an effective method for someone other than the operator to promptly supervise the success of tube placement on the opposite side.

While the use of a video laryngoscope contributes to an increased success rate of intubation, success is not always guaranteed and is contingent on the operator's proficiency and the patient's level of airway difficulty. In our study, video bronchoscopy was employed for all intubations except one, and in two cases where video bronchoscopy was utilized, esophageal intubation occurred. It is important to note that our study did not focus on comparing the accuracy between TTUS and Video Laryngoscopy. Instead, our objective was to compare the effectiveness of TTUS for endotracheal tube (ETT) confirmation in CPR situations involving the use of PAPR with that of EtCO2. The limitation of not considering various ETT confirmation methods simultaneously was acknowledged in the Limitation Section (lines 414-417).

“Forth, tube placement should be checked using multiple methods, as the simultaneous use of two or more independent tests improves the overall sensitivity. However, several commonly used clinical signs were not evaluated in this study.”

This study represents the first exploration of tracheal intubation with TTUS while wearing a PAPR during CPR for cardiac arrest patients. While some studies have compared the accuracy of ultrasound and capnography in confirming tube placement in a cardiac arrest setting, none have undertaken a direct comparison based on the visualization of endotracheal intubation through the glottis in CPR environments with PAPR. Therefore, our research contributes to filling a previously unexplored gap in existing studies, investigating the effectiveness of ultrasound for verifying ETT placement in CPR situations involving the use of PAPR.

We have revised the entire manuscript in a tone and manner that emphasizes this point.

The title has been updated as follows:

"Real-time Tracheal Ultrasound vs. Capnography for Intubation Confirmation during CPR with a Powered Air-Purifying Respirator in the COVID-19 Era".

Some sentences in the abstract have been modified as follows:

“Introduction: This study aimed to compare the accuracy of real-time trans-tracheal ultrasound (TTUS) with capnography to confirm intubation in cardiopulmonary resuscitation (CPR) while wearing a powered air-purifying respirator (PAPR). This setting reflects increased caution due to contagious diseases.

Conclusion: This study demonstrated that TTUS is more accurate and faster than EtCO2 measurement for confirming endotracheal tube placement during CPR, particularly in the context of PAPR usage in pandemic conditions.”

We revised some sentences in the introduction as follows (Lines 61-66) (Lines 86-88):

“During the COVID-19 pandemic, it is recommended for physicians performing intubations to wear personal protective equipment (PPE), including powered air-purifying respirators (PAPR). This requirement not only makes auscultation impractical but also could impair vision due to the PAPR hood. Consequently, this may increase the complexity of the intubation process”

“Therefore, this research investigates the effectiveness of ultrasound for verifying ETT placement in CPR situations involving the use of PAPR, filling a previously unexplored gap in existing studies.”

It was emphasized in the Discussion section as follows (Lines 322-337)

“Confirmation of tracheal tube placement is important, particularly in the COVID-19 era. Falsely placed tubes harm not only the patient but also expose healthcare providers to the potential risk of a contagious disease.[23] Wearing PPE by medical staff, common in the COVID-19 era, creates various physical restrictions and raises the necessity to verify whether the methods used to confirm tube placement still function while wearing PPE. This is the first study to determine tracheal intubation with TTUS while wearing a PAPR during CPR for cardiac arrest patients. Although the number of participants was modest, TTUS demonstrates to be a very useful procedure with 100% accuracy for tube confirmation. Moreover, the median time for con-firming intubation by TTUS was only 12 s, which was significantly faster than the time to EtCO2 measurement of 45 s. Consequently, the performance of TTUS was as good as that of studies in which medical staff did not wear PPE;[1,24] it is considered a useful approach to con-firm intubation during CPR while wearing PAPR during the pandemic era.”

We revised some sentences in the conclusion as follows (Lines 426-429)

“This study demonstrates that TTUS is a more effective method than capnography for confirming endotracheal tube placement during CPR, particularly when healthcare providers are equipped with PAPRs in pandemic situations.”

This is the first study to determine tracheal intubation with TTUS while wearing a PAPR during CPR for cardiac arrest patients. We have updated our manuscript to better emphasize these novel and crucial aspects of our research during the pandemic. Although it does not present a new perspective on the usefulness of tracheal ultrasound, it can be said to have newly presented its usefulness in CPR while wearing PPE in the special situation of the pandemic era. Your constructive feedback is instrumental in refining our paper and enhancing its contribution to the field.

Round 2

Reviewer 1 Report

Comments and Suggestions for Authors

Thank yo very much for having addressed my comments. Now the quality of the paper has increased. I don't have any major issues.

Reviewer 2 Report

Comments and Suggestions for Authors

The authors respond adequately to the reviewers' questions, considering that it can be published in its current version.